# Thinning sea ice weakens buttressing force of iceberg mélange and promotes calving

Alexander A. Robel[1,2]

At many marine-terminating glaciers, the breakup of mélange, a floating aggregation of sea ice and icebergs, has been accompanied by an increase in iceberg calving and ice mass loss. Previous studies have argued that mélange may suppress calving by exerting a buttressing force directly on the glacier terminus. In this study, I adapt a discrete element model to explicitly simulate mélange as a cohesive granular material. Simulations show that mélange laden with thick landfast sea ice produces enough resistance to shut down calving at the terminus. When sea ice within mélange thins, the buttressing force on the terminus is reduced and calving is more likely to occur. When a calving event does occur, it initiates a propagating jamming wave within mélange, which causes local compression and then slow mélange expansion. The jamming wave can also initiate widespread fracture of sea ice and further increase the likelihood of subsequent calving events.

[1] Division of Geological and Planetary Sciences, California Institute of Technology, Pasadena, California 91125, USA. [2] Department of the Geophysical Sciences, University of Chicago, Chicago, 60637 Illinois, USA. Correspondence and requests for materials should be addressed to A.A.R. (email: robel@caltech.edu).

Projections of the future ice sheet contribution to sea level rise can vary by several metres depending on the rate of iceberg calving at ice sheet margins[1–3]. The recent increase in the rate of iceberg calving and associated mass loss at many marine-terminating glaciers in Greenland and Antarctica has been accompanied by rapid breakup of iceberg mélange, a dense aggregation of icebergs and sea ice floating in front of glacier termini[4–6]. Calving at some glaciers in Greenland shuts down each winter when sea ice within mélange is thick and rapidly resumes in spring coincident with sea ice breakup[7,8]. These co-occurrences suggest that mélange may suppress calving through the direct application of force on the glacier terminus[5,9,10].

Model simulations of outlet glaciers[11–15] have shown that iceberg calving can be suppressed entirely by prescribing a sufficiently large mélange buttressing force at a glacier terminus. The observed seasonal variation in calving and terminus position at several marine-terminating outlet glaciers in Greenland can be reproduced by prescribing this mélange buttressing force in winter and removing it in summer. Furthermore, in some models[12,15], mélange buttressing is found to have a significant effect on not just the seasonality of calving, but also the annually averaged calving rate, terminus position and glacier velocity near the terminus. Previous studies have used tidal variations in hydrostatic pressure at the terminus as an analogue to estimate the magnitude of this mélange buttressing force and its seasonal and longer-term evolution[10,14]. However, the physical processes that control the magnitude of this mélange buttressing force and its seasonal and longer-term evolution are not well understood, because existing continuum ice flow models are not capable of explicitly simulating the interactions between discrete icebergs and sea ice floes that occur within mélange. To understand the complex two-way interaction that occurs between mélange and glaciers requires a novel approach to modelling mélange.

In this study, I adapt the Discrete-Element bonded-particle Sea Ice model[16], a toolbox of the open-source Large-scale Atomic/Molecular Massively Parallel Simulator Improved for General Granular Heat Transfer Simulations[17] to simulate mélange as a cohesive granular material. The adapted model calculates the motions and interactions of a large population of cylindrical icebergs floating in seawater and bonded by elastic plates of sea ice that dynamically break and re-form. Force is transmitted within the mélange through direct contact between icebergs and elastic deformation of sea-ice bonds. Icebergs also experience form and skin drag from ocean water, which is assumed to be at rest. Iceberg diameters are randomly drawn from a truncated log-normal distribution derived from observations[18]. The parameterizations associated with these model simplifications are not well constrained by the limited set of mélange observations that exist, but the results discussed in this study are not sensitively dependent on these parameter choices (Table 1).

Using this new approach to modelling mélange, I show that when bonded by a matrix of thick landfast sea ice, mélange is capable of exerting sufficient buttressing force on the glacier terminus to shut down calving. Thinning of sea ice reduces mélange buttressing force on the terminus, increasing the likelihood of a calving event occurring. When a calving event does occur, it initiates a propagating jamming wave within the mélange, which can cause fracture of the sea-ice matrix. Such large-scale sea ice fracture within mélange decreases the buttressing force of mélange on the glacier terminus and further increases the likelihood of subsequent calving events.

## Results

**Mélange buttressing force and sea ice thickness.** In winter months, sea ice in Greenland fjords can grow to many metres in

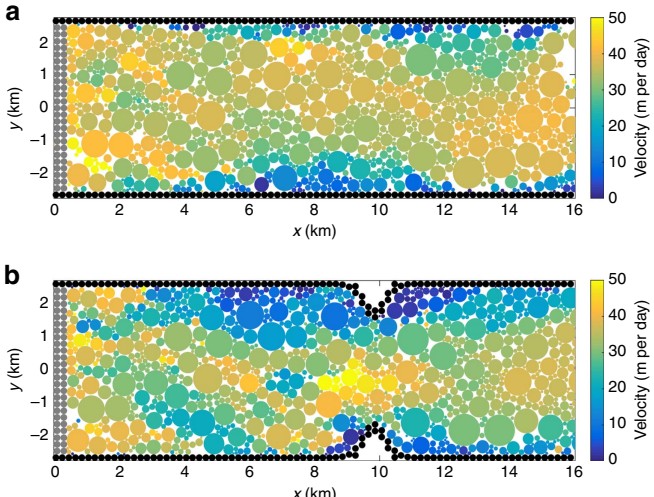

**Figure 1 | Snapshots of iceberg positions and velocities in two different channel configurations.** Both snapshots are 11 days into simulations with steady terminus advance and no calving, with 2-m-thick sea ice bonds, which are not plotted here. Velocity of each iceberg element is indicated by filled colour. Terminus elements are filled grey circles. Channel sides are filled black circles. (**a**) Straight channel configuration. (**b**) Narrowed channel configuration.

**Table 1 | Parameters used in this study.**

| Parameter | Description | Value |
|---|---|---|
| $C_w$ | Dimensionless drag coefficient for icebergs in seawater | 0.01 |
| $E_b$ | Sea ice elastic modulus (Pa) | $9 \times 10^9$ |
| $E_i$ | Iceberg elastic modulus (Pa) | $9 \times 10^9$ |
| $r_{min}$ | Minimum iceberg radius (m) | 10 |
| $r_{max}$ | Maximum iceberg radius (m) | 500 |
| $r_\mu$ | Log-normal median of iceberg radius (m) | 90 |
| $r_\sigma$ | Log-normal standard deviation of iceberg radius (m) | 117 |
| $\Delta t$ | Time step (s) | 0.002 |
| $\lambda_b$ | Bond length coefficient | 0.01 |
| $\lambda_R$ | Bond radius coefficient | 1.0 |
| $\lambda_{ns}$ | Bond ratio of normal to shear stiffness | 1.5 |
| $\mu$ | Static yield criterion | 0.7 |
| $\nu$ | Iceberg Poisson's Ratio | 0.33 |
| $\rho_i$ | Iceberg density (kg m$^{-3}$) | 910 |
| $\rho_w$ | Seawater density (kg m$^{-3}$) | 1028 |
| $\sigma_{c,max}$ | Bond compressive strength (Pa) | $2 \times 10^6$ |
| $\sigma_{t,max}$ | Bond tensile strength (Pa) | $1 \times 10^6$ |
| $\tau_{max}$ | Bond shear strength (Pa) | $5 \times 10^5$ |

thickness due to the freezing-in of small pieces of icebergs and strong pressure ridging[19,20]. In a series of simulations, I vary the prescribed thickness of sea ice to determine the influence of sea-ice state on the time- and width-averaged buttressing force exerted by mélange on an advancing terminus.

I consider two idealized channel configurations that resemble many fjords in Greenland. The 'straight' channel configuration (Fig. 1a) has a uniform width of 5 km along its length. Narrowing in channel geometry has also been hypothesized to play a role in the dynamics of mélange in the Illulisat Icefjord in which Jakobshavn Isbræ terminates, through the formation of ice dams during the winter[7]. To test the importance of these geometric effects, I also consider a 'narrowed' channel configuration, which incorporates a gaussian narrowing of the channel to 3 km at a location 10 km down-fjord from the initial terminus position

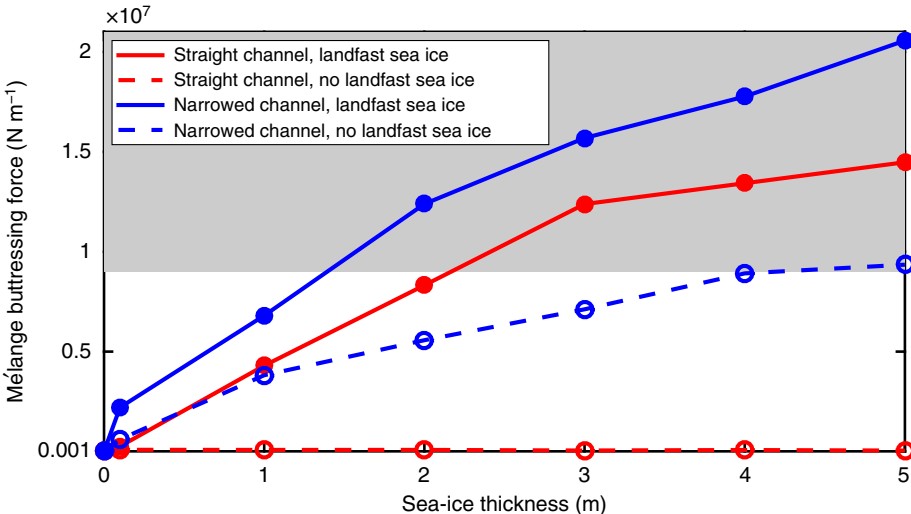

**Figure 2 | Mélange buttressing force on the terminus as a function of sea-ice bond thickness.** Buttressing force is obtained by averaging over the terminus width and over 1 month of simulations run to a statistical steady state. Red lines are simulations with a straight channel configuration. Blue lines are simulations with a narrowed channel configuration. Solid lines are simulations that permit sea-ice bonds between mélange elements and between mélange elements and channel side elements. Dashed lines are simulations with sea-ice bonds only permitted between mélange elements. Grey shading indicates the mélange buttressing force that prevents penetration of fractures near terminus, as estimated in Fig. 9 of Krug et al.[15] and consistent with the Amundson et al.[9] estimates for prevention of iceberg rotation from a terminus at flotation (for $\theta = 1°$).

(Fig. 1b). The channel sides are composed of static elements, while the terminus is composed of elements prescribed to moved along the channel length at 40 m per day.

Figure 2 shows that in all model configurations mélange buttressing force is an increasing function of sea-ice thickness. Thicker sea ice transmits more stress through the mélange and is more resistant to stress-induced breaking. Some force on the terminus originates from friction within the mélange and between mélange and channel sides. Most of the force comes from landfast sea ice (simulated in the model as bonds formed between iceberg elements and channel side elements), which is strongly sheared over the narrow region between icebergs and channel sides (see Supplementary Movie 1 showing evolution of mélange simulated with landfast sea ice). In simulations with thick sea ice inside mélange, but none between icebergs and channel sides (dashed lines in Fig. 2), the force on the terminus is always more than 40% lower than in the equivalent simulations with landfast sea ice (solid lines). Sea ice becomes landfast in fjords or embayments where it is pushed into land boundaries and not easily exported, which is generally the case in locations where mélange tends to aggregate[20]. Though Fig. 2 includes simulations without landfast sea ice to illustrate the important role of landfast sea ice, those simulations that include landfast sea ice (solid lines) can be considered to be the more realistic representations of channelized or embayed glaciers in front of which mélange aggregates.

In the narrowed channel configuration (blue lines in Fig. 2), the mélange buttressing force is always higher than in the straight channel configuration (red lines in Fig. 2). Contact and sea ice deformation between icebergs and channel sides oriented across the primary direction of mélange motion contributes significantly to compressive forces within the mélange upstream of the narrowing. The aggregate result is akin to a granular Bernoulli effect[21], where mélange flow accelerates through the constricted channel (see snapshot of mélange velocity in Fig. 1b), and pressure is higher in the region upstream of the narrowing. When sea ice is sufficiently thick, this higher pressure is more effectively transmitted upstream to the terminus.

Previous studies have estimated that a width-averaged mélange buttressing force of order $10^7$ N m$^{-1}$ will shut down calving by:

(1) preventing rotation of fractured icebergs away from the terminus[9] and (2) suppressing crevasse penetration near the terminus that prevents iceberg detachment in the first place[15]. When sea ice within mélange (including landfast sea ice) is sufficiently thick, the simulated magnitude of mélange buttressing force exerted on the terminus is within this range that is sufficient to suppress iceberg calving (grey shading in Fig. 2). In a narrowed channel, this suppression may occur at lower sea ice thickness. This magnitude of buttressing force is also consistent with studies that reproduce the observed seasonal variation in calving rate and terminus position of Jakobshavn Isbræ by prescribing a seasonal variation in buttressing pressure at the terminus of depth-integrated glacier models[11,12]. Here I provide the first plausible physical verification that wintertime thickening of sea ice within fjords can cause sufficient strengthening of mélange to shut down calving. As a corollary, these simulations demonstrate that the thinning of sea ice within mélange reduces force on the terminus to levels that would allow vigorous calving activity to occur.

**Calving events and jamming waves.** From late spring until early fall, mélange at the terminus of many marine-terminating outlet glaciers in Greenland primarily consists of icebergs with little-to-no interstitial sea ice. During this period, observations indicate that mélange typically moves at the same speed as the glacier terminus, punctuated by calving events that initiate short intervals of rapid down-fjord mélange motion at speeds thousands of times faster than the background rate of advance[22–24]. These events flush icebergs out of the fjord and so provide a critical control on terminus–mélange contact and the delivery of icebergs and freshwater to the global ocean.

I simulate a summertime calving event by prescribing rapid forward motion of a 1,000 m-wide cluster of iceberg elements, at 2.5 m s$^{-1}$, away from the terminus over a period of 5 min in the narrowed channel configuration. Figure 3 (and Supplementary Movie 2) shows the rapid spatial (Fig. 3a) and temporal (Fig. 3b) response of mélange simulated without sea ice. This response constitutes a 'jamming wave'[24,25], which is initiated when the

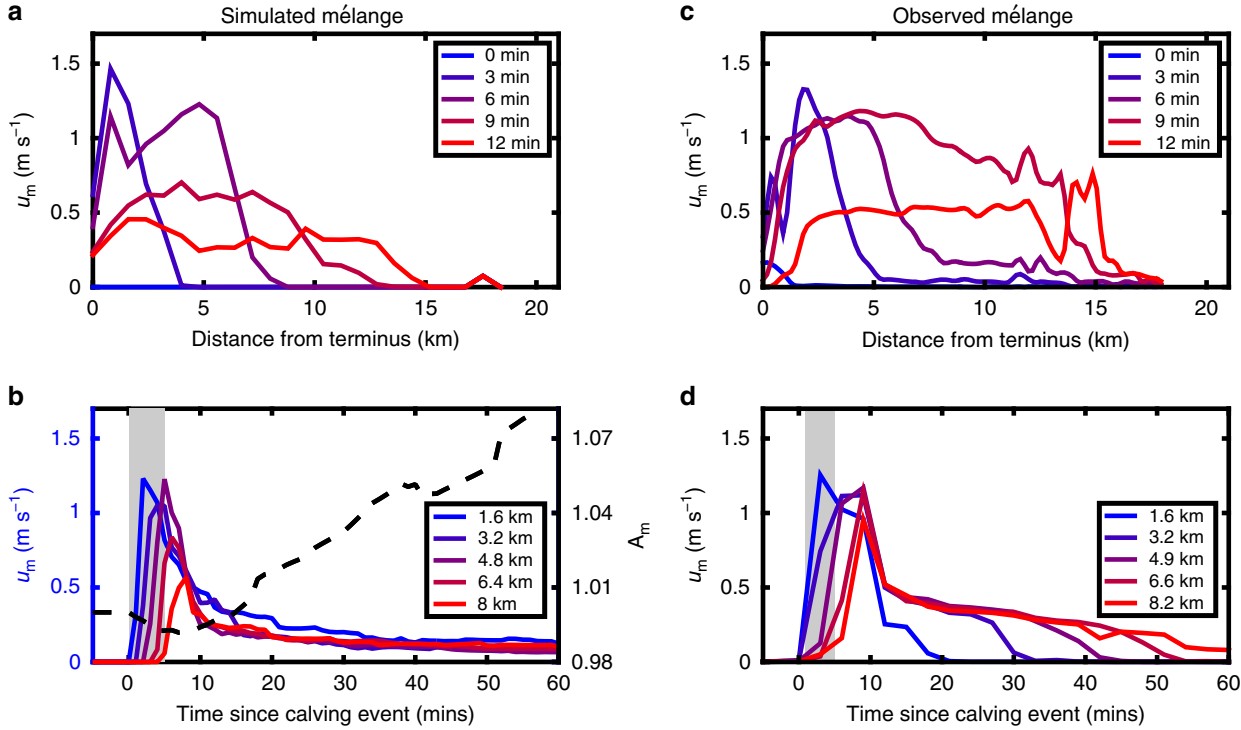

**Figure 3 | Spatiotemporal response of mélange without sea ice to a prescribed calving event.** (**a,b**) are simulations, and (**c,d**) are observations from Jakobshavn Isbræ reproduced from Fig. 3 in Peters *et al.*[24]. (**a,c**) Mélange velocity (averaged across channel width) as a function of distance along channel length at different times following calving event. (**b,d**) Mélange velocity (averaged across channel width) as a function of time following calving event at different locations along channel length. Grey shading indicates duration of calving event. Black dashed line plots the area occupied by mélange normalized by pre-calving area.

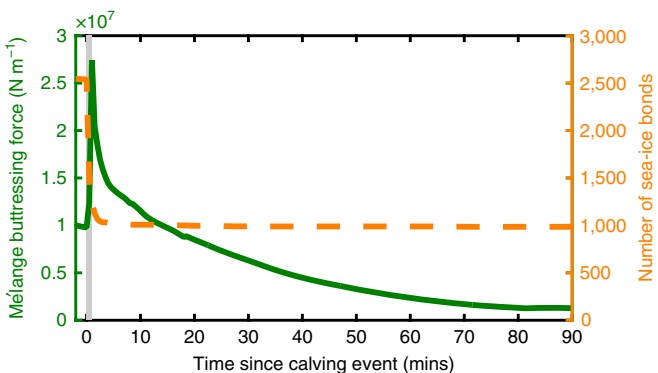

**Figure 4 | Simulated response of mélange with 2-m-thick sea ice to a calving event.** Green solid line is mélange buttressing force (averaged over terminus width and not including calving iceberg). Orange dashed line is the number of sea-ice bonds as a function of time following calving event.

calved iceberg makes contact with other icebergs near the terminus, which then make contact with other nearby icebergs in a propagating wave of locally compressed (jammed) mélange. For comparison, measurements of mélange speed following a calving event in Illulisat Icefjord (determined using particle image velocimetry processing of time-lapse photography and ground-based interferometry, described in Peters *et al.*[24]) are reproduced in Fig. 3c,d. The average speed (20–30 m s$^{-1}$) and width (1–3 km) of the simulated jamming wave front are broadly similar to these observations, with some small differences in the rate of mélange slowdown following the passage of the jamming wave. Otherwise, individual iceberg speeds ($\sim$1 m s$^{-1}$) are comparable to iceberg-based GPS measurements that capture rapid mélange flushing events[22]. The jamming wave causes

compression of mélange throughout the channel in the 10 min following the calving initiation and then a slow expansion over several hours (black dashed line in Fig. 3b). Over the next few days, mélange slowly returns to the background rate of glacier advance as it is recompressed by the glacier terminus, in agreement with satellite-derived observations[23].

**Spring sea ice breakup and the resumption of calving.** The vigorous calving at Jakobshavn in summer is in marked contrast to the complete lack of calving in winter. A rapid transition between these two states occurs in the spring, when sea ice within the mélange begins to slowly thin as incoming shortwave radiation increases and air temperatures warm. Several weeks before air temperatures in the fjord exceed freezing, sea ice rapidly breaks up in a period of hours to days[7,26]. Vigorous calving then resumes, and the short floating ice tongue that has formed during the winter cessation of calving quickly disintegrates, leaving a grounded terminus.

To understand the mechanism and speed of this spring transition, I simulate the first major calving event occurring in the midst of slow spring thinning of sea ice within mélange. Mélange in the narrowed channel configuration is initialized with sea-ice bonds of 2 m thickness and then brought to a statistical steady state through slow terminus advance where the buttressing force at the terminus is in the range associated with the resumption of calving (grey shading in Fig. 2). I initiate a small calving event by prescribing the forward motion of a 500 m-wide ice element, at 0.5 m s$^{-1}$, away from the terminus over a period of 1 min. Figure 4 shows the force exerted on the terminus, not including the calved iceberg, by mélange (green solid line) and the number of sea-ice bonds present within the mélange (orange dashed line) in the time surrounding this prescribed calving event. Within a

few minutes of the calving event, a jamming/elastic wave propagates quickly though the mélange by contact between icebergs and elastic compression of sea ice (Supplementary Movie 3). Approximately 60% of the sea ice in the channel is fractured by the passage of this wave. The initial compression of mélange near the terminus causes a brief period of increased buttressing force, which then drops by over 80% from the pre-calving state, as mélange slowly expands away from the terminus. In the region of the terminus directly surrounding the initial calving event, mélange is no longer in contact with the terminus at all. Though recompression of mélange and sea-ice refreezing may slowly raise the buttressing force over a period of days, the lowered force greatly increases the likelihood that progressively larger calving events will occur within hours after the first calving event during the expansion of mélange. Such a cascade of progressively larger calving events is initiated by slow changes in sea ice during spring, but it will rapidly lead to persistently sea-ice-free mélange and a summer-like mélange state with vigorous calving.

## Discussion

In this study, I use a new modelling approach to simulate the interaction of icebergs and sea ice within mélange. Idealized simulations show that, in general, iceberg mélange laden with thick landfast sea ice exerts a buttressing force on an advancing glacier terminus which is in a quantitative range that has been shown to suppress calving in previous modelling studies. The simulated magnitude of mélange buttressing force is based on the actual physical processes which occur within mélange, and not tidal analogues as in the previous studies[10,14]. Simulations show that the buttressing force is principally the result of resistance to shearing along channel sidewalls by landfast sea ice. In scenarios where the channel narrows downstream of the terminus, compression within mélange increases the magnitude of buttressing force at the terminus and reduces the thickness of sea ice necessary to suppress calving. Joughin et al.[7] have suggested that such a channel narrowing (or shallowing) may play a role in the formation of an 'ice dam' of mélange down-fjord of Jakobshavn Isbræ during mid-winter. However, even in fjords without such a narrowing, it is possible that mélange buttressing force may be sufficient to suppress calving, given sufficiently thick sea ice formation in winter. The idealized nature of these simulations and the uncertainty associated with the material parameters of mélange make it difficult to directly apply the quantitative model predictions in this study to specific marine-terminating glaciers. However, as future observations provide better constraints on the properties of mélange, this model can be used with more realistic fjord geometry to estimate the mélange buttressing force at Jakobshavn Isbræ and other marine-terminating glaciers in Greenland and Antarctica.

The dependence of simulated mélange buttressing force on sea ice thickness suggests that interannual changes in sea ice seasonality and thickness may lead to two possible outcomes. The first possible outcome is a change in the seasonality of calving rate, but no overall change in mean annual calving rate. That is, as winter sea ice thins, more calving occurs in winter, but less calving occurs in summer. Mean annual calving rate is controlled, in part, by the large-scale glacier geometry and velocity near the terminus[27], which may be uneffected by changes in calving on short timescales. However, it may also be possible that sea ice loss leads to less seasonality in calving rate, but also an increase in annual mean calving rate, and hence a retreat of the terminus. This outcome acknowledges the possibility that the processes that control individual calving events and calving rate over short time scales may have some influence on long-term calving rates as

well. The model I use does not explicitly simulate the fracture processes that lead to calving, and so on its own cannot determine which of these two possible outcomes is likely to occur in reality. However, previous studies, which force iceberg calving models with a prescribed seasonal fluctuation in mélange buttressing force of a similar magnitude to that simulated here[12,15], find that if either the duration or magnitude of the winter mélange buttressing force is decreased, the result is an increase in annual mean calving rate and retreat of the annually averaged position of the terminus. Krug et al.[15] suggest that the loss of mélange buttressing leads to calving on short time scales, which then causes higher glacier velocities at the terminus due to decreased lateral shear stress and a torque at the vertical glacier cliff face that persist on long time scales. In conjunction with these studies, I conclude that the likely effect of shorter and thinner winter sea ice within mélange is an increase in mean annual glacier mass loss. In the future, these approaches can be combined by explicitly modelling mélange and iceberg calving simultaneously.

In the model described in this study, sea ice bonds are removed from a simulations after they reach a threshold in strength and break. In reality, when sea ice breaks due to shear or tensile failure, it is still physically present and may have compressive strength greater than zero. Thus, the model may underestimate the transmission of compressive stresses due to the omission of broken sea ice, and hence the buttressing force that mélange exerts on the terminus. However, since compression tends to transmit the least amount of stress across sea ice bonds, this error due to the lack of compression is likely a small fraction of the total stress exerted on a given iceberg.

When sea ice becomes thinner, mélange produces a lower buttressing force on the terminus, which may permit calving to occur. Simulated calving events initiate the propagation of jamming waves within iceberg mélange that cause brief periods where icebergs move at speeds in excess of $1\,\mathrm{m\,s^{-1}}$ and many icebergs leave the fjord system. The simulated jamming waves qualitatively reproduce jamming waves observed in Illulisat Icefjord, though decay of iceberg velocities following passage of the jamming wave is faster than in reality. These differences may arise due to the use of simple quadratic drag law for resistance from ocean water, though variation of the drag coefficient over a range of realistic values does not strongly change the qualitative character of simulations. Jamming waves and their decay are more strongly controlled by the contact between icebergs. It is likely the case that mismatch between simulations and observations is due to the use of a rate-independent Coulomb friction law to simulate contact between icebergs, whereas laboratory experiments show that the coefficient of friction for ice exhibits velocity dependence[28]. One intriguing possibility is that detailed observations of jamming waves in mélange (such as Peters et al.[24]) can be used to constrain the frictional properties of glacial ice in real-world settings, which is important for understanding fracture and crevassing in ice sheets.

When an iceberg calves into mélange laden with sea ice, a jamming/elastic wave rapidly propagates down-fjord causing compression, fracture of sea ice and a decrease in mélange buttressing force at the terminus. The buttressing force is then sufficiently low that subsequent calving events are more likely to occur. If the rate of sea ice refreezing and mélange compression by terminus advance is sufficiently slow compared to the frequency of subsequent calving events, then a cascade of calving events will rapidly fracture all sea ice within the mélange and lead to a state of vigorous calving. As Amundson et al.[9] have suggested, the ice just upstream of a glacier terminus prevented from calving and overturning by thick winter sea ice in mélange may be heavily fractured and prone to rapid calving upon release of the mélange buttressing force. The subsequent cascade of

calving events may explain the rapid breakup of mélange and resumption of calving at Jakobshavn Isbræ that takes place over a few days each spring[7]. Even in Antarctica, where there is persistent, landfast, multi-year sea ice at the termini of some outlet glaciers, the transition from quiescence to a state of vigorous calving has been observed to occur over a similarly short time period of a few days, in concert with sea ice breakup[6].

If the rate of iceberg calving from the Greenland and Antarctic Ice Sheets accelerates, sea level rise may also accelerate to rates that could reach several metres per century over the next several centuries[1–3]. Ocean and atmosphere warming in the Arctic have already caused longer periods of sea-ice-free conditions[29], which may lead to a transition from currently slow calving, predominantly occurring in the summer, to rapid calving, occurring throughout the year. Conversely, increased runoff from the ice sheet and iceberg melting in fjords may create a layer of cold, fresh water promoting sea ice formation[30]. The distribution of meltwater depends critically on the rate of iceberg export from the fjord and subsequent advection and melting, which may lead to widespread cooling over Greenland[31]. Including mélange in models that predict the evolution of ice sheets and sea level is critical to capturing the many interacting processes that cause complicated feedbacks and potentially unanticipated future changes.

## Methods

**Iceberg mélange model.** The Discrete-Element bonded-particle Sea Ice model (DESIgn)[16] is a toolbox of the open-source Large-scale Atomic/Molecular Massively Parallel Simulator Improved for General Granular Heat Transfer Simulations (LAMMPS-LIGGGHTS)[17]. Iceberg interactions are simulated using a classical Hertzian model for elastic contact between disks with a Coulomb friction law and weak viscoelastic damping to maintain stability.

Sea-ice bonds are elastic plates with width, $R_{ij} = \lambda_R \min(r_i, r_j)$, length, $b_{ij} = \lambda_b(r_i + r_j)$, and a prescribed thickness, where $r$ is the radius of the two bonded icebergs. I choose $\lambda_b$ to be small, as there should be minimal overlap in sea ice on icebergs (rather icebergs are 'frozen in' to interstitial sea ice) and most sea-ice deformation should occur in a narrow zone between icebergs. A sea-ice bond is created when two icebergs without an existing bond come within 10 m of one another. A sea-ice bond is broken, and removed from the simulation, when either the tensile, compressive or shear stresses on the bond exceed material strength thresholds for ice. The material properties of sea ice in mélange, which is composed of a mix of bergy bits and *in situ* sea ice with significant pressure ridging, are not well known. Consequently, I use strength thresholds falling in the middle of the experimentally measured range for old sea ice, reviewed in Timco and Weeks[32], including compressive strength that tends to be higher than tensile strength.

**Iceberg size distribution.** The iceberg size distribution is a truncated log-normal distribution based on ship-based observations of iceberg size on the Greenland continental shelf[18]. I shift the observed iceberg distribution to higher diameters to account for the melting that occurs in the time between the calving of an iceberg from a glacier and the measurement of iceberg sizes on the continental shelf. Icebergs smaller than 10 m are considered to be sea ice floes, and so are not included in the simulations. Icebergs larger than 1 km can present potential issues with motion through the channel and so are also disallowed. Iceberg thickness is set to the iceberg diameter.

**Construction of the initial mélange state.** Iceberg diameters are randomly drawn from the distribution described above. Icebergs are then placed randomly within the channel and then compressed to a desired packing density. In the summer jamming wave simulations (Fig. 3a,b), a buffer zone of loosely packed icebergs is added down-fjord of the primary mélange (with an initially prescribed packing fraction near a jammed state). This buffer zone prevents rapid ejection of icebergs at the edge of the main mélange zone, and will arise naturally in simulations of mélange run to steady state (such as those in Fig. 2, see Supplementary Movie 1), in addition to actual summertime mélange.

**Data availability.** All code written by the author and used to run these simulations is available freely as a public GitHub repository at https://github.com/aarobel/melange-dem. LAMMPS-LIGGGHTS is a software package available publicly from the CFDEM project (http://cfdem.com) through an open-source licence. The DESIgn toolbox is available from the website of A. Herman (http://herman.ocean.ug.edu.pl/LIGGGHTSseaice.html).

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

## Acknowledgements

I thank D. MacAyeal and D. Abbot for comments on the manuscript, V. Tsai, J. Amundson, J. Burton, B. Lipovsky, C. Horvat and W. Zhang for discussions, and I. Peters for providing the data. A. Herman provided timely and very helpful model support. All model simulations were performed on the Midway High Performance Cluster at the University of Chicago. Financial support was provided by the NOAA Climate and Global Change Postdoctoral Fellowship administered by the University Corporation for Atmospheric Research and the Caltech Stanback Postdoctoral Fellowship.

## Author contributions

A.A.R. designed the study, devised and executed the mélange simulations, analysed the simulations and wrote the manuscript.

## Additional information

**Competing financial interests:** The author declares no competing financial interests.

**Publisher's note**: 

