## [Peer Review File · Nature Communications]

Reviewers' comments:

Reviewer #1 (Remarks to the Author):

This manuscript details a process study of the buttressing influence provided by iceberg melange, the dense aggregate of icebergs and sea ice, on tidewater glaciers. A granular model is applied to represent the coupled influence of discrete icebergs frozen together by a matrix of sea ice. The diameter of the icebergs are varied according to a distribution inferred from observations. Model runs are investigated over a range of sea ice thicknesses, for two different idealized fjord geometries, and for lateral boundary conditions representing whether or not the sea ice is frozen to the fjord margins. A glacier terminus advancing into the fjord is represented as a rigid translation of the inlet boundary. Similar model runs investigate the impact of a discrete iceberg calving event impacting the melange. The primary result of the model runs is the resultant buttressing force on the glacier terminus. The writing in the manuscript is clear, although I expected a bit more context related to the link between buttressing and larger scale ice dynamics and calving. The primary conclusion alluded to in the title, that thinning sea ice reduces buttressing, is rather intuitive and not especially revealing.

This study puts quantitative bounds on the appropriate level of melange buttressing force depending on the fjord geometry and sea ice thickness. Previous studies have used wider ranges of potential buttressing force for applying in an ice sheet modeling context, and this study is indeed novel in that it explicitly models melange dynamics in order to better constrain the amount of buttressing for different scenarios. This is a welcome development for wider scale studies that include explicit glacier and calving dynamics (which this study does not).

This study basically confirms the widespread intuition that iceberg melange bonded by a matrix of sea ice should apply some buttressing force to a tidewater glacier, with the force scaling with the thickness of the sea ice. The results indicate that, for the most part, the level of buttressing is not sufficient to suppress calving unless the fjord has a constriction AND the sea ice is especially thick AND there is landfast sea ice on the fjord margins. It would be worthwhile to discuss how commonly these conditions are fulfilled, otherwise the implications of this study are not clear. Illulisat is the only example mentioned that might fulfill these conditions, but how common are they more generally for outlet glaciers? The abstract mentions that melange is "capable of exerting sufficient buttressing force... to shut down calving" but if this is only for very special conditions in only a few places, then the results are not so consequential (in which case there isn't much of a "wow" factor to justify publication in such a high-impact journal).

The manuscript is framed by a statement (lines 9-11) that ice sheet models cannot test the hypothesis that melange suppresses calving by directly applying force to the glacier terminus. However, this has been done, even by one of the studies referenced in the manuscript (Krug et al. 2015; see also Todd and Christoffersen "Are seasonal calving dynamics forced by buttressing from ice melange or undercutting by melting? Outcomes from full-stokes simulations of Store Glacier, West Greenland." *Cryosphere* 8, 2353–2365 (2014)). These studies specify a buttressing force as a boundary condition on the glacier front in an ice sheet model, with explicit parametrizations for calving. In my opinion, the present study does not adequately recognize these studies nor place the results in an appropriate context with respect to the insights gleaned from these studies. This may just be a matter of re-framing the appropriate context of this study, but as it reads it sounds a bit like the author is discounting (or unaware of) previous studies of buttressing and calving using ice sheet models.

Ice shelves are discussed at several points in the paper (in the abstract and the concluding paragraph), with the implication that the present study addresses buttressing in this context as well. However, calving from ice shelves commonly occurs after rift propagation that separates large, tabular icebergs. This style of calving operates over very different spatial and temporal scales than calving from a tidewater glacier in fjord filled with melange. The model scenarios in

this study do not address these scales. I suggest removing reference to ice shelves in the manuscript, as this discussion is out of context and somewhat misleading.

Reviewer #2 (Remarks to the Author):

In this paper the author uses a discrete particle model to estimate the strength of a proglacial melange. A series of papers have postulated that this melange could influence calving or perhaps provide sufficient resistance to influence glacier flow. But none of the previous work has been able to show whether the necessary forces were of a magnitude that could reasonably be accounted for by an iceberg/sea ice mixture. This is the first study to quantify this. It fills an important gap in the understanding of melange/glacier interaction. The paper is very well written and it will make a significant contribution. It can basically be published as is.

Two details:

l.7: I believe the correct spelling is 'termini'

l.35/36: Can you specify what the small end of the ice berg distribution looks like? I assume that the Dowdeswell observations don't contain measurements of the small brash ice that is common in glacial fjords. Below a certain size class this might just be considered part of the sea ice, but it would be helpful to explicitly state what was done here.

Martin Truffer

Reviewer #3 (Remarks to the Author):

A. Summary of the key results

In the manuscript, a discrete-element model is used to simulate the dynamics of a mélange of sea ice and icebergs in a narrow fjord in front of a calving glacier. A series of idealized simulations is performed to demonstrate that thick sea ice filling spaces between icebergs – typical for winter and early spring conditions – can exert on the glacier terminus forces sufficient to suppress calving. Further, the simulations suggest that, when the ice is thin and therefore weak, fracturing of sea ice associated with a post-calving “shock” wave propagating along the fjord tends to weaken the ice further, and thus to facilitate subsequent calving events. These results are supported by the existing (though limited) observations, cited in the text.

B. Originality and interest

To the best of my knowledge, the work presented in this manuscript is the first application of discrete-element techniques to modeling of sea ice/iceberg mélange.

The results suggest that the state of the sea ice cover in front of glaciers terminating in the sea may have substantial influence on calving rates and thus – presumably – on mass balance of those glaciers. High seasonal variability of sea ice thickness, with thick ice in winter and thin or no ice in the summer, favors short periods of extremely high calving rates in spring, following a “calm” winter of gradually advancing glacial front, when no calving occurs. On the other hand, lower sea ice thickness throughout a year would mean more uniform distribution of calving events in time, as the mélange buttressing force wouldn't be sufficient to prevent calving even in winter. A big question is how these two “regimes” may influence the total, annual average volume of glacier ice “lost” for calving. In this way, the results of this manuscript provide a very interesting starting point for further research, relevant in the context of the changing climate and the role of the cryosphere in that process.

The manuscript also demonstrates the applicability of discrete-element models to relevant problems in cryospheric research.

C. Data & methodology: validity of approach, quality of data, quality of presentation

The work is based on modeling with a discrete-element model called DESIgn. The model configuration, described in the "Methods Summary" (and briefly in the main text), is presented in a clear way. The detailed comments given below, related to technical aspects of the model setup, do not affect my generally very positive opinion regarding this aspect of the manuscript.

1. The author doesn't mention how the interactions of the model icebergs with the underlying ocean are treated. And no related model parameters are listed in Table 1. Is the ocean at rest? Does it have a prescribed velocity? Are both drag components (skin + form drag) used? For icebergs (which in the model setup used have thicknesses identical to their diameters) the form drag is substantial and this choice should have a very strong influence on, e.g., the reaction of icebergs to the jamming wave after calving.

2. A consequence of the fact that sea ice is represented in the model as elastic bonds connecting icebergs is that, when it breaks, it simply disappears and its strength drops instantaneously to zero. When real sea ice is broken, it obviously is still there and, even if its tensile strength may be zero, its compressive strength remains larger from zero. I think that a few sentences regarding possible artificial effects related to this model property would be helpful.

3. In line 210: R_{ij} is bond width and b_{ij} – the bond length, not the other way round (in Table 1, the naming is correct).

4. In the model description, the author mentions that there are two mechanisms that may lead to bond breaking. The second, distance-based one, is clearly not physical. Has the author checked if (and how often) this second mechanism actually does lead to bond breaking in the simulations performed? Why the first, stress-based mechanism, is not enough?

5. Table 1 suggests that the same compressive and tensile strength is used for sea ice. I don't suppose this could affect the conclusions of the simulations, but it is unrealistic.

D. Appropriate use of statistics and treatment of uncertainties

does not apply

E. Conclusions: robustness, validity, reliability

Conclusions are valid and supported by the results presented.

One comment: In my opinion, it is disputable whether thin ice in winter would *necessarily* mean higher total volumes of glacier ice lost for calving, as the author suggests. Another possibility is that the volume (at longer time scales) would stay roughly the same, but instead of a clear maximum of calving rates in spring, their distribution would be more uniform in time. In other words, in "thick-ice regime" the spring calving maximum simply removes what had accumulated in winter, and in "thin-ice regime" calving occurs throughout winter as the ice front gradually advances. Is there observational evidence that justifies the "thinner sea ice -> higher annual-mean calving rates" conclusion? Or any theoretical arguments favoring it?

F. Suggested improvements: experiments, data for possible revision

None.

G. References: appropriate credit to previous work?

Yes.

H. Clarity and context: lucidity of abstract/summary, appropriateness of abstract, introduction and conclusions

The manuscript, including its abstract, is written in a very clear, readable way.

Reviewer #1 (Remarks to the Author):

This manuscript details a process study of the buttressing influence provided by iceberg melange, the dense aggregate of icebergs and sea ice, on tidewater glaciers. A granular model is applied to represent the coupled influence of discrete icebergs frozen together by a matrix of sea ice. The diameter of the icebergs are varied according to a distribution inferred from observations. Model runs are investigated over a range of sea ice thicknesses, for two different idealized fjord geometries, and for lateral boundary conditions representing whether or not the sea ice is frozen to the fjord margins. A glacier terminus advancing into the fjord is represented as a rigid translation of the inlet boundary. Similar model runs investigate the impact of a discrete iceberg calving event impacting the melange. The primary result of the model runs is the resultant buttressing force on the glacier terminus. The writing in the manuscript is clear, although I expected a bit more context related to the link between buttressing and larger scale ice dynamics and calving. The primary conclusion alluded to in the title, that thinning sea ice reduces buttressing, is rather intuitive and not especially revealing.

I see where the reviewer is coming from here, which I believe is connected to the suggestion (of reviewers 1 and 3) to place these results in the context of previous studies which connect the buttressing force of melange to calving rates. I have added “and promotes calving” to the manuscript title to make this extra link more clear, and (as I discuss more below), I have added additional discussion (in the introduction and discussion sections) of the existing work linking melange buttressing force to calving and mass loss on seasonal and longer time scales. These changes make the larger point that changes in the buttressing force provided by melange are important to calving on both short and long time scales.

This study puts quantitative bounds on the appropriate level of melange buttressing force depending on the fjord geometry and sea ice thickness. Previous studies have used wider ranges of potential buttressing force for applying in an ice sheet modeling context, and this study is indeed novel in that it explicitly models melange dynamics in order to better constrain the amount of buttressing for different scenarios. This is a welcome development for wider scale studies that include explicit glacier and calving dynamics (which this study does not).

This study basically confirms the widespread intuition that iceberg melange bonded by a matrix of sea ice should apply some buttressing force to a tidewater glacier, with the force scaling with the thickness of the sea ice. The results indicate that, for the most part, the level of buttressing is not sufficient to suppress calving unless the fjord has a constriction AND the sea ice is especially thick AND there is landfast sea ice on the fjord margins. It would be worthwhile to discuss how

commonly these conditions are fulfilled, otherwise the implications of this study are not clear. Illulisat is the only example mentioned that might fulfill these conditions, but how common are they more generally for outlet glaciers? The abstract mentions that melange is “capable of exerting sufficient buttressing force? to shut down calving” but if this is only for very special conditions in only a few places, then the results are not so consequential (in which case there isn’t much of a “wow” factor to justify publication in such a high-impact journal).

This perceptive point shows me that I have not properly explained the purpose of the various different types of simulations in Figure 2 in the original draft. I aim to communicate that, broadly, the magnitude of the buttressing force for melange with thick sea ice is within the range that previous studies have estimated necessary for suppressing calving.

Landfast sea ice is a general feature of fjords and embayments with significant sea ice cover and especially in locations where there is a dense aggregation of icebergs. So, in general we expect landfast ice to form in places where melange is forming. Rather, these simulations are simply meant to show that landfast sea ice is one of the root mechanisms producing the buttressing stress. This point is now made explicitly in the new draft.

Reviewer 3 made an excellent point that the distance breaking criteria was unphysical. Indeed, this criteria was causing entirely too much sea ice breaking in simulations for unphysical. Based on this suggestion, I removed the distance breaking criterion and lowered the shear strength of ice to a level in the middle of what is expected from laboratory and field measurements. The results are somewhat stronger back forces in the straight channel case, which mostly resolves this issue of calving suppression in straight vs. narrowed channels. Sufficiently thick inter-iceberg sea ice produces high buttressing stresses even in the absence of fjord narrowing.

Given the idealized nature of these simulations and the parametric uncertainty associated with the properties of ice melange, my purpose is not to focus on where exactly the lines in Figure 2 cross the lower boundary of the gray shading, which is based on estimates from simulations which are themselves idealized and subject to parametric uncertainty. This study should be seen as outlining the most important factors which cause the buttressing stress magnitude of the right order to suppress calving, given the current state of knowledge about melange. It also provides a guide to future studies about what properties of melange are the most important to constrain through observations and then to model correctly when focusing on specific outlet glaciers.

All of these points have now been made in explicit detail in the results and discussion sections.

The manuscript is framed by a statement (lines 9-11) that ice sheet models cannot test the hypothesis that melange suppresses calving by directly applying force to the glacier terminus. However, this has been done, even by one of the studies referenced in the manuscript (Krug et al. 2015; see also Todd and Christoffersen “Are seasonal calving dynamics forced by buttressing from

ice melange or undercutting by melting? Outcomes from full-stokes simulations of Store Glacier, West Greenland.” *Cryosphere* 8, 2353-2365 (2014)). These studies specify a buttressing force as a boundary condition on the glacier front in an ice sheet model, with explicit parametrizations for calving. In my opinion, the present study does not adequately recognize these studies nor place the results in an appropriate context with respect to the insights gleaned from these studies. This may just be a matter of re-framing the appropriate context of this study, but as it reads it sounds a bit like the author is discounting (or unaware of) previous studies of buttressing and calving using ice sheet models.

Thank you to the reviewer for this excellent point. I have rewritten this part with considerably more detail to make it clear that previous studies have tested this idea by exploring the influence of a prescribed melange back force on calving rates. This study seeks to understand the physical processes which control the magnitude of the melange back force by explicitly modeling melange, which is generally not possible in continuum ice sheet flow models. I didn't intend to discount previous studies, and this new text (in the introduction and conclusion) should place my study within the appropriate context of previous modeling.

Ice shelves are discussed at several points in the paper (in the abstract and the concluding paragraph), with the implication that the present study addresses buttressing in this context as well. However, calving from ice shelves commonly occurs after rift propagation that separates large, tabular icebergs. This style of calving operates over very different spatial and temporal scales than calving from a tidewater glacier in fjord filled with melange. The model scenarios in this study do not address these scales. I suggest removing reference to ice shelves in the manuscript, as this discussion is out of context and somewhat misleading.

This point is well taken. I have removed the references to ice shelf collapse associated with sea ice loss in the introduction. I have also added a short discussion of this distinction between rotational calving and rift calving in the discussion section.

Reviewer #2 (Remarks to the Author):

In this paper the author uses a discrete particle model to estimate the strength of a proglacial melange. A series of papers have postulated that this melange could influence calving or perhaps provide sufficient resistance to influence glacier flow. But none of the previous work has been able to show whether the necessary forces were of a magnitude that could reasonably be accounted for by an iceberg/sea ice mixture. This is the first study to quantify this. It fills an important gap in the understanding of melange/glacier interaction. The paper is very well written and it will make a significant contribution. It can basically be published as is.

Two details:

l.7: I believe the correct spelling is 'termini'

Fixed

l.35/36: Can you specify what the small end of the iceberg distribution looks like? I assume that the Dowdeswell observations don't contain measurements of the small brash ice that is common in glacial fjords. Below a certain size class this might just be considered part of the sea ice, but it would be helpful to explicitly state what was done here.

I have added more discussion of the modified iceberg size distribution to methods section, in particular my reasons for truncating the size distribution at low iceberg diameters. As you note, below a certain size class, icebergs may be considered part of the sea ice, and so I do not include icebergs less than ten meters in diameter, which is the threshold size for creation of sea ice bonds.

Martin Truffer

Reviewer #3 (Remarks to the Author):

A. Summary of the key results

In the manuscript, a discrete-element model is used to simulate the dynamics of a mlang of sea ice and icebergs in a narrow fjord in front of a calving glacier. A series of idealized simulations is performed to demonstrate that thick sea ice filling spaces between icebergs typical for winter and early spring conditions can exert on the glacier terminus forces sufficient to suppress calving. Further, the simulations suggest that, when the ice is thin and therefore weak, fracturing of sea ice associated with a post-calving "shock" wave propagating along the fjord tends to weaken the ice further, and thus to facilitate subsequent calving events. These results are supported by the existing (though limited) observations, cited in the text.

B. Originality and interest

To the best of my knowledge, the work presented in this manuscript is the first application of discrete-element techniques to modeling of sea ice/iceberg mlang. The results suggest that the state of the sea ice cover in front of glaciers terminating in the sea may have substantial influence on calving rates and thus presumably on mass balance of those glaciers. High seasonal variability of sea ice thickness, with thick ice in winter and thin or no ice in the summer, favors short periods of extremely high calving rates in spring, following a "calm" winter of gradually advancing glacial front, when no calving occurs. On the other hand, lower sea ice thickness throughout a year would mean more uniform distribution of calving events in time, as the mlang buttressing force wouldn't be sufficient to prevent calving even in winter. A big question is how these two

regimes may influence the total, annual average volume of glacier ice lost for calving. In this way, the results of this manuscript provide a very interesting starting point for further research, relevant in the context of the changing climate and the role of the cryosphere in that process. The manuscript also demonstrates the applicability of discrete-element models to relevant problems in cryospheric research.

C. Data & methodology: validity of approach, quality of data, quality of presentation

The work is based on modeling with a discrete-element model called DESIgn. The model configuration, described in the Methods Summary (and briefly in the main text), is presented in a clear way. The detailed comments given below, related to technical aspects of the model setup, do not affect my generally very positive opinion regarding this aspect of the manuscript. 1. The author doesn't mention how the interactions of the model icebergs with the underlying ocean are treated. And no related model parameters are listed in Table 1. Is the ocean at rest? Does it have a prescribed velocity? Are both drag components (skin + form drag) used? For icebergs (which in the model setup used have thicknesses identical to their diameters) the form drag is substantial and this choice should have a very strong influence on, e.g., the reaction of icebergs to the jamming wave after calving.

The description of the drag law was omitted from the original manuscript due to space constraints, but I have added it since space constraints are no longer an issue with Nature Communications. The ocean, which is at rest, does indeed exert a skin and form drag on icebergs in the simulations. As is typically done, this drag force is parameterized as a quadratic function. The strength of the drag appears to have little influence on the jamming wave (short of unphysical high drag coefficients), since contact between icebergs is the dominant force causing motion during jamming waves. I've added these points in the discussion of the jamming wave simulations.

The longer version of this argument is that, assuming a high drag coefficient (for a cylinder in water at high Reynolds number), the force associated with drag of a large iceberg moving at 1 m/s through water is approximately: $0.5 * C_d * \rho * A * (v^2) = 0.5 * 0.01 * 1000 * (1000^2) * (1^2) = 0.005$ gN, whereas the average contact force between icebergs is more like 1 gN. Thus, for the absolute largest iceberg going at max speed with a high drag coefficient, this drag force may be important after icebergs are out of contact, but for icebergs in contact and any smaller icebergs, the drag force will not significantly change the results, except maybe to make the decay faster at high drag coefficient, which would make the simulations match less well to observations.

2. A consequence of the fact that sea ice is represented in the model as elastic bonds connecting

icebergs is that, when it breaks, it simply disappears and its strength drops instantaneously to zero. When real sea ice is broken, it obviously is still there and, even if its tensile strength may be zero, its compressive strength remains larger from zero. I think that a few sentences regarding possible artificial effects related to this model property would be helpful.

This is an excellent point. We have added in a discussion of the potential underestimate of stresses that this may introduce.

3. In line 210: R_{ij} is bond width and b_{ij} the bond length, not the other way round (in Table 1, the naming is correct).

Yes, this is correct. Error has been fixed in text.

4. In the model description, the author mentions that there are two mechanisms that may lead to bond breaking. The second, distance-based one, is clearly not physical. Has the author checked if (and how often) this second mechanism actually does lead to bond breaking in the simulations performed? Why the first, stress-based mechanism, is not enough?

This was a really nice point that got me to go back and re-examine some simulations. Essentially, I had put this distance breaking criterion in place originally to ensure that bonds would not stick around too long as icebergs drifted apart. What I found is that by decreasing the shear strength of sea ice bonds (to a level that falls in the middle of the range expected from laboratory and field measurements), this distance breaking condition is not necessary anymore - shear breaking works just fine in eliminating bonds. So, it has been removed, and all the simulations have been updated accordingly. It does seem to be the case that this distance breaking condition was entirely too aggressive at breaking landfast sea ice bonds before, and so the number of sea ice bonds and back force of the melange on the terminus is somewhat higher in the landfast sea ice simulations in these new simulations. This does not qualitatively change any conclusions of this study, but it does resolve an question raised by reviewer 1 (whether calving suppression only occurs in special fjord cases - it does not).

5. Table 1 suggests that the same compressive and tensile strength is used for sea ice. I don't suppose this could affect the conclusions of the simulations, but it is unrealistic.

This is a good point. I originally made all the strength thresholds the same because I reasoned that the uncertainty on these values makes it difficult to justify having one value be different from the others. That being said, as you have pointed out, the literature (particularly the excellent review by Timco and Weeks in Cold Regions Science and Tech. from 2010) seems to agree that sea ice compressive strengths tend to be higher than tensile strengths. In fact, 1 MPa is quite a low value for compressive strength, so I have changed it to 2 MPa in these new simulations. The

results change a little numerically, but not qualitatively.

D. Appropriate use of statistics and treatment of uncertainties

does not apply

E. Conclusions: robustness, validity, reliability

Conclusions are valid and supported by the results presented.

One comment: In my opinion, it is disputable whether thin ice in winter would necessarily mean higher total volumes of glacier ice lost for calving, as the author suggests. Another possibility is that the volume (at longer time scales) would stay roughly the same, but instead of a clear maximum of calving rates in spring, their distribution would be more uniform in time. In other words, in “thick-ice regime” the spring calving maximum simply removes what had accumulated in winter, and in “thin-ice regime” calving occurs throughout winter as the ice front gradually advances. Is there observational evidence that justifies the “thinner sea ice -> higher annual-mean calving rates” conclusion? Or any theoretical arguments favoring it?

This reviewer raises a very good point here, and one that I had not previously acknowledged in the discussion. The discussion section does now include an explanation that distinguishes between two possible outcomes of lower duration/magnitude of winter melange buttressing force. One is a change in seasonality without a change in annual mean calving rate (the possibility raised by this reviewer). The other is a change in seasonality and a change in annual mean calving rate. In two studies (Vieli and Nick 2011; Krug et al. 2015) where they have prescribed such changes in seasonality (both duration and magnitude) of melange buttressing stress, they find that annual mean calving rates increase and the annual mean terminus position retreats. The theory of Krug et al. 2015 is that the loss of melange buttressing leads to calving events, which causes higher glacier velocities at the terminus due to decreased lateral shear stress (lower contact area at side walls) and a torque at the vertical glacier cliff. These two studies use different calving models of differing complexity, and so provide somewhat independent lines of evidence that there is an annual-mean effect of changing melange seasonality. This discussion is included in the discussion section.

F. Suggested improvements: experiments, data for possible revision

None.

G. References: appropriate credit to previous work?

Yes.

H. Clarity and context: lucidity of abstract/summary, appropriateness of abstract, introduction and conclusions

The manuscript, including its abstract, is written in a very clear, readable way.

REVIEWERS' COMMENTS:

Reviewer #1 (Remarks to the Author):

The revised manuscript is much improved, and I am very impressed with the detailed response by the author to each reviewer's comments. I believe that this work, as presented now, will be a welcome contribution to the glaciological community. The manuscript is well written and referenced, and I have no further major comments.

The only minor comment I have relates to a comment I made in the original submission, and this is the attempted link between this melange buttressing work and the subject of ice shelf buttressing in Antarctica. Although some revision was made based on my original remarks, the Discussion section still attempts to place the present work in a context that is relevant to tabular calving, ice shelf buttressing, and even ice shelf collapse (!). Previous work related to the presence and influence of melange between the flanks of ice shelf rifts may be tangentially related to the present work (lines 196-202 in revised text).

However, lines 237-242 implies that reduced sea ice may actually be causally related to the loss of ice shelves and resultant ice shelf buttressing in Antarctica ("... if the Antarctic Ice Sheet loses buttressing... Such a scenario may occur as sea ice melts..."). The implied causal link seems to be that sea ice/melange has a controlling role in ice shelf stability, which is counter to prevailing notions of ice shelf stability. Since this is quite a speculative and unsubstantiated statement, I would repeat my original criticism here that the link to Antarctic ice shelf buttressing is not warranted. Rift propagation and tabular calving from ice shelves is, to first order, governed by the prevailing glaciological stresses in ice shelves. Other links have been made between rift propagation/calving and tsunami arrival times from major earthquakes. However, sea ice buttressing has not been shown to have a major role in ice shelf stability or ice shelf buttressing. The implications of the present study are indeed wide enough without this attempted link to ice shelves. I would again suggest that the author remove this tenuous link.

Reviewer #2 (Remarks to the Author):

I had very few comments initially, and I think this manuscript is ready for publication and constitutes a really nice contribution.

Reviewer #3 (Remarks to the Author):

In my opinion, the revised paper is suitable for publication without further modifications. The Author has very convincingly replied to all comments and suggestions, and the new version of the paper has been significantly improved (especially the discussion part).

Reviewer #1 (Remarks to the Author):

The revised manuscript is much improved, and I am very impressed with the detailed response by the author to each reviewer's comments. I believe that this work, as presented now, will be a welcome contribution to the glaciological community. The manuscript is well written and referenced, and I have no further major comments.

The only minor comment I have relates to a comment I made in the original submission, and this is the attempted link between this melange buttressing work and the subject of ice shelf buttressing in Antarctica. Although some revision was made based on my original remarks, the Discussion section still attempts to place the present work in a context that is relevant to tabular calving, ice shelf buttressing, and even ice shelf collapse (!). Previous work related to the presence and influence of melange between the flanks of ice shelf rifts may be tangentially related to the present work (lines 196-202 in revised text).

However, lines 237-242 implies that reduced sea ice may actually be causally related to the loss of ice shelves and resultant ice shelf buttressing in Antarctica (“... if the Antarctic Ice Sheet loses buttressing... Such a scenario may occur as sea ice melts...”). The implied causal link seems to be that sea ice/melange has a controlling role in ice shelf stability, which is counter to prevailing notions of ice shelf stability. Since this is quite a speculative and unsubstantiated statement, I would repeat my original criticism here that the link to Antarctic ice shelf buttressing is not warranted. Rift propagation and tabular calving from ice shelves is, to first order, governed by the prevailing glaciological stresses in ice shelves. Other links have been made between rift propagation/calving and tsunami arrival times from major earthquakes. However, sea ice buttressing has not been shown to have a major role in ice shelf stability or ice shelf buttressing. The implications of the present study are indeed wide enough without this attempted link to ice shelves. I would again suggest that the author remove this tenuous link.

I see how this set of sentences can be confusing. I was mostly attempting to accurately represent the conditional nature of the DeConto & Pollard (2016) results, but bringing in ice shelf collapse does confuse the issue somewhat. I have re-written this paragraph, and removed the entirety of the other paragraph in the discussion section on shelf rifting (which I do not strictly need to address here). The new paragraph simply gets across the points that iceberg discharge at grounded termini is important to sea level rise. The rate and seasonality of iceberg calving may be changed by changing sea ice cover. This sticks directly to the results of the paper, without going too far afield. More broadly, I have now removed all mentions of ice shelf collapse from the manuscript.

Reviewer #2 (Remarks to the Author):

I had very few comments initially, and I think this manuscript is ready for publication and constitutes a really nice contribution.

Reviewer #3 (Remarks to the Author):

In my opinion, the revised paper is suitable for publication without further modifications. The Author has very convincingly replied to all comments and suggestions, and the new version of the paper has been significantly improved (especially the discussion part).